# Depletion of the Origin Recognition Complex Subunits Delays Aging in Budding Yeast

**DOI:** 10.3390/cells11081252

**Published:** 2022-04-07

**Authors:** Karolina Stępień, Adrianna Skoneczna, Monika Kula-Maximenko, Łukasz Jurczyk, Mateusz Mołoń

**Affiliations:** 1Department of Biology, Institute of Biology and Biotechnology, University of Rzeszów, 35-601 Rzeszów, Poland; karolina.stepien89@interia.pl; 2Institute of Biochemistry and Biophysics, Polish Academy of Sciences, 02-106 Warsaw, Poland; 3The Franciszek Górski Institute of Plant Physiology, Polish Academy of Sciences, 30-239 Krakow, Poland; m.kula@ifr-pan.edu.pl; 4Institute of Agricultural Sciences, Land Management and Environmental Protection, University of Rzeszow, 35-601 Rzeszów, Poland; ljurczyk@ur.edu.pl

**Keywords:** aging, cell cycle, lifespan, ORC, replication, yeast

## Abstract

Precise DNA replication is pivotal for ensuring the accurate inheritance of genetic information. To avoid genetic instability, each DNA fragment needs to be amplified only once per cell cycle. DNA replication in eukaryotes starts with the binding of the origin recognition complex (ORC) to the origins of DNA replication. The genes encoding ORC subunits have been conserved across eukaryotic evolution and are essential for the initiation of DNA replication. In this study, we conducted an extensive physiological and aging-dependent analysis of heterozygous cells lacking one copy of *ORC* genes in the BY4743 background. Cells with only one copy of the *ORC* genes showed a significant decrease in the level of *ORC* mRNA, a delay in the G1 phase of the cell cycle, and an extended doubling time. Here, we also show that the reducing the levels of Orc1-6 proteins significantly extends both the budding and average chronological lifespans. Heterozygous *ORC/orcΔ* and wild-type diploid cells easily undergo haploidization during chronological aging. This ploidy shift might be related to nutrient starvation or the inability to survive under stress conditions. A Raman spectroscopy analysis helped us to strengthen the hypothesis of the importance of lipid metabolism and homeostasis in aging.

## 1. Introduction

Replication is a cellular metabolic process in which a cell duplicates one or more DNA molecules. DNA replication begins when the origins of replication are recognized and bound by initiation proteins and is a tightly regulated process ensuring that genetic information is inherited. DNA replication initiation is stringently regulated and involves multiple layers of backup mechanisms [1]. Replication is initiated from the origins of replication recognized and bound by the origin recognition complex (ORC) [2,3,4]. The ORC is a six-subunit ATP-dependent DNA-binding protein encoded by *ORC1-6* [5]. The first origin recognition complex proteins were identified in budding yeast as a heterohexameric complex that specifically bound to the origins of DNA replication [2]. Orthologs of *ORC1-5* were identified in divergent organisms from fruit flies to humans, which greatly suggested that these genes were strongly evolutionarily conserved [6,7]. In turn, *ORC6* genes showed no structural similarity to the other ORC proteins, and were poorly conserved between budding yeast and other eukaryotic species [7]. The role of the ORC is to form a scaffold for a number of additional replication factors in the G1 phase of the cell cycle, including Cdc6, Cdt1 and Mcm2-7, which together form a pre-replication complex [2,8,9,10,11]. The ORC remains bound to chromatin at replication origins throughout the cell cycle [4], but is only active in the late mitosis and early G1 phase of the cell cycle. Both Orc1p and Orc5p bind ATP, though only Orc1p has ATPase activity [12]. The Orc5p and Orc6p are crucial for the stability of ORC as the whole structure/complex [13]. Yeast ORC is a human ORC orthologue; therefore, yeast may serve as a model in aging research and during studies of humanized yeast used to reveal the mechanism leading to human diseases, to which defects in ORC functioning contribute. So far, the mutations in inter alia *ORC1*, *ORC4* and *ORC6* were linked to dwarfism, also known as Meier-Gorlin syndrome [14]. In this paper, we focused on the effect of reducing the levels of Orc1-6 proteins on cellular aging. In general, aging is defined as a multifactorial process leading to the loss of function of cells, tissues and organs. Yeast has been frequently and successfully used in aging research for both aspects of this process, in both mitotically active [15] and post-mitotic cells [16]. There are two main approaches to aging used in the studies employing yeast as the model system. The classical approach to replication aging examines the proliferative potential of cells, expressed as the ability of the yeast mother cell to produce daughter cells [17]. This method was later enhanced by the cell’s or organism’s lifetime parameters (as in other models) [18] and is a model for studying active mitotic cells in higher eukaryotes [19]. In the second approach, the so-called chronological aging is used as a model for studying post-mitotic cells of higher eukaryotes, including humans [16,20]. Many factors are known to influence the aging of yeast, including calory restriction [21], anti-aging compounds [22,23,24], ribosome biogenesis disorders [25,26,27], mitochondrial dysfunction [28], and an increased level of reactive oxygen species, leading to oxidative damage [29] or extrachromosomal rDNA circles (ERCs) formation [30,31]. However, despite extensive research on aging, little is known about the possible linkage between replication initiation and aging. This is hardly strange considering that most of the factors involved in this process are essential for cell viability. In our studies, we took advantage of the Orc1-6 protein level difference between cells that were homozygous or heterozygous with regard to *ORC* genes. We asked if decreased levels of proteins, which were necessary for replication to start, e.g., Orc1-6 proteins, would affect the aging process of these cells. We showed that a lowered level of any of the Orc1-6 proteins resulted in a significant increase in the budding lifespan and delayed the average chronological aging, likely due to the delay in the cell cycle and subsequent extension of the doubling time.

## 2. Materials and Methods

### 2.1. Strains and Growth Conditions

All yeast strains used in this study are listed in Table 1.

Yeast cells were grown in a standard liquid-rich YPD medium (1% Difco Yeast Extract, 1% Yeast Bacto-Peptone, and 2% glucose) on a rotary shaker at 150 rpm or on solid rich YPD medium containing 2% agar. The experiments were performed at 28 °C.

### 2.2. Determination of the Growth Rate

The growth assays were carried out on liquid medium as described above. The yeast cell suspension was incubated for 12 h with shaking at 28 °C (Heidolph Incubator 1000 at 1200 rpm). The growth was monitored in the Anthos 2010 type 17,550 microplate reader at 600 nm by measurements at 2 h intervals for 12 h. In the second approach, the number of cells per ml in each culture was counted using a Malassez chamber (Carl Roth, Lauda-Konigshofen, Germany).

### 2.3. Determination of the Mean Doubling Time

The mean doubling time was calculated for each analyzed strain as described previously [32]. The doubling time was calculated during the routine determination of the budding lifespan. The times of the first two reproductive cycles were not taken into account (the first and second doubling times are longer than those of older cells). The data represents mean values from three independent experiments (with 45 cells used in each experiment) with a mean standard deviation (SD). Statistically significant differences were taken at a *p*-value of < 0.001 using the one-way ANOVA.

### 2.4. Cell Cycle Analysis

The cells were grown to OD_600_ 0.3–0.5, and 1 mL of cell culture was harvested by centrifugation, and then washed with water fixed with a chilled (−20 °C) 70% ethanol (Polmos, Warsaw, Poland) for 2 h at room temperature. The fixed cells were then washed twice in FACS buffer (0.2 MTris-HCl (Sigma-Aldrich, Burlington, MA, USA), pH 7.4, 20 mM EDTA (Merck, Darmstadt, Germany), and incubated for 2 h at 37 °C in FACS buffer with 1 mg per ml RNase A (Sigma-Aldrich, Burlington, MA, USA) to eliminate RNA. The cells were then washed with phosphate-buffered saline (PBS) and stained overnight at 4 °C in the dark with 100 μL of propidium iodide solution (50 μg per ml in PBS; Calbiochem, San Diego, CA, USA). After the addition of 900 μL PBS, the cells were sonicated three times per 10 s in an ultrasonic bath, Branson 2800 (Branson Ultrasonic Corporation, Danbury, CT, USA), prior to FACS analysis of the DNA content. Analysis was performed using a FACS Calibur analyzer (Becton-Dickinson, Franklin Lakes, NJ, USA). A total of 10,000 cells were counted in a single assay. The mean percent of cells in G1, S, and G2 phases of the cell cycle was counted from three independent biological replicative. The standard deviation was also calculated. The usage of two control strains, haploid and diploid, enabled the gating of cells in the respective cell cycle phase.

### 2.5. Measurement of Cell Metabolic Activity

Metabolic activity of yeast cells was determined with FUN-1 according to the manufacturer’s protocol (Molecular Probes, Eugene, OR, USA) with small modifications described in [33]. The fluorescence of the cell suspension was measured after 15 min incubation in the dark and at 28 °C using the TECAN Infinite 200 microplate reader (Grodig, Austria) at λ_ex_ = 480 nm and λ_em_ = 500–650 nm. The metabolic activity of cells was expressed as a change in ratio of red (λ = 575 nm) to green fluorescence (λ = 535 nm). The mean and SD were calculated from the data of at least four cultures for each strain.

### 2.6. Detection of Rad52-YFP Foci via Fluorescence Microscopy

To determine the number of Rad52-YFP foci, *ORC* deletion strains and BY4743 wild type were used. The yeast strains were transformed with the plasmid pWJ1344 carrying a RAD52-YFP fusion [34]. The transformants were grown in SC-Leu medium at 28 °C with shaking until the exponential phase. An aliquot of each culture was collected for microscopic analysis, which allowed the frequency of the spontaneous formation of Rad52 foci to be assessed. The remaining culture was treated with 1 mM H_2_O_2_ for one hour, then aliquots of cell suspensions were taken for a microscopic analysis of a further quantification of the frequency of oxidative stress-induced Rad52 foci. The cells were examined under a fluorescence microscope (Olympus BX51, Tokyo, Germany). The number of cells and Rad52-YFP foci in the cells were counted, and the percentage of cells with Rad52-YFP foci and the average number of Rad52-YFP foci per cell were calculated after screening of at least 500 cells.

### 2.7. Sporulation Efficiency Assay

Diploid strains pre-grown in rich YPD medium were placed onto sporulation medium containing 0.1% yeast extract, 1% potassium acetate, 0.05% glucose, 2% agar for 14 days at 28 °C, as previously described [35]. Cells were then suspended in water. Cells and asci were counted in a cell-counting chamber Malassez (at least 300 cells per probe), and the frequency of asci among the total cells was expressed as a percentage of asci relative to all cells counted and WT. The mean and SD were calculated from the data for at least three cultures of each strain.

### 2.8. Tetrad Dissection

Tetrad dissection was performed as previously described [35]. Cells were placed onto sporulation medium containing 0.1% yeast extract, 1% potassium acetate, 0.05% glucose, and 2% agar for 14 days at 28 °C. Then, cells were suspended in 0.5 mg/mL Zymolyase 100T solution in 50 mM Tris-HCl pH 7.5, and incubated for 10 min at 37 °C. Tetrads were dissected using a Nikon Eclipse E200 optical microscope with an attached micromanipulator on rich YPD plates. Spores were grown for three days at 23 °C.

### 2.9. RNA Isolation and RT-qPCR

For the extraction of mRNA, the yeast RiboPure RNA Purification Kit (Invitrogen, Waltham, MA, USA; AM1926) was used in accordance with the manufacturer’s instructions. For mRNA extraction, 2 × 10^7^ cells were collected from overnight cultures. The quality and yield of the RNA were checked using the TECAN Infinite 200 microplate reader. The samples were stored at a concentration of 300 µg/µL in 5 mg aliquots at −80 °C.

The quality of samples was determined by analyzing the RNA concentration (260 nm) and its purity (280/230 nm) in a 20-fold diluted solution of each sample in nuclease-free water on a UV-1800 spectrometer (Shimadzu, Tokyo, Japan). The isolates had an average concentration of 918.933 ng/mL (σ = 167.178). To remove residual dsDNA, ssDNA and RNA–DNA complexes, samples were incubated for 30 min at 37 °C (TBD-100, Biosan, Riga, Latvia) with 50 U RNase-free DNAase (10 U/µL, A&A Biotechnology, Gdynia, Poland) in buffer containing 50 mM Tris-HCl (pH = 8.0) and 5 mM MgCl2. After 5 min incubation, the reaction was stopped by heating at 75 °C, followed by cooling to 4 °C.

The reverse transcription was performed in two steps. The synthesis of first strand was performed using the TranScriba Kit (A&A Biotechnology, Gdynia, Poland) in MasterCycler^®^ thermocycler (Eppendorf, Hamburg, Germany) as follows: an average of 1026 µg (σ = 0.015) of each RNA sample was incubated with 1 µL of oligo(dT)18 primer at 65 °C for 5 min. in the total volume of 9.5 µL (nuclease-free water), then chilled down to 4 °C. After adding a reaction mixture consisting of 4 µL of 5× reaction buffer, 0.5 µL of RNase inhibitor (40 U/µL), 2 µL of dNTP (10 mM) and 4 µL of recombinant reverse transcriptase MMLV (20 U/µL) filled with sterile water up to 20 µL, the reaction was conducted at 42 °C for 60 min, then stopped at 70 °C for 5 min, and the resulting cDNA were stored at −20 °C until further analysis. In the second step, the relative number of transcripts was measured by quantitative PCR in Light Cycler^®^ 96 instrument (Roche, Basel, Switzerland). The reaction mixture consisted of: 7.5 µL RT PCR Mix SYBR^®^ A (Taq 0.1 U/μL, MgCl_2_ 4 mM, 0.5 mM of each dNTP in 2× reaction buffer containing SYBR^®^ Green, A&A Biotechnology, Gdynia, Poland), 0.56 µL of each primer (*ORC* and *ACT1* to normalize the data; 10 mM, Genomed, Warsaw, Poland, for details see Table 2), 0.75 µL of cDNA (previously diluted 1:5 in sterile water), all filled up to 15 µL with sterile water. The reaction was carried out with the following thermal profile: 180 s of pre-denaturation at 95 °C, followed by 40 cycles comprising denaturation at 95 °C for 30 s, annealing at 55 °C for 30 s, and an extension at 72 °C for 30 s with a single acquisition. Each reaction was completed via a melting analysis of the products with the following thermal profile: 95 °C for 10 s, 65 °C for 60 s, then up to 97 °C with 5 readings/°C. Each sample was processed in at least two biological and three technical replicates, and PCR reactions were conducted in the presence of a negative reagent control and a positive control with yeast DNA. Potential contamination of RNA isolates by DNA was excluded by preliminary PCR with ACT1 primers, which was then separated on horizontal gel electrophoresis. The entire analysis was carried out in accordance with the principles of good practice in RNA handling.

### 2.10. Determination of Budding Lifespan

After overnight growth, the cells were arrayed on a YPD plate using a micromanipulator. The budding lifespan was microscopically determined by a routine procedure with the use of a micromanipulator, as described previously [36]. The number of buds formed by each mother cell reflected its reproductive potential (budding lifespan). In each experiment, 45 single cells were analyzed. The results represented measurements for at least 90 cells analyzed in two independent experiments. The analysis was performed by micromanipulation using the Nikon Eclipse E200 optical microscope (Nikon, Amsterdam, Netherlands) with an attached micromanipulator.

### 2.11. Determination of the Total Lifespan

The total lifespan is the length of life of a single mother yeast cell expressed in units of time and was calculated as the sum of reproductive (time between the first and the last budding) and post-reproductive lifespans (time from the last budding until cell death). The total lifespan of the *S. cerevisiae* yeast was determined as previously described in [18] with small modifications from [36]. Ten-microliter aliquots of a fresh exponential culture of yeast were collected and transferred on YPD plates with solid medium containing Phloxine B (10 μg/mL). In each experiment, 45 single cells were analyzed. During manipulation, the plates were kept at 28 °C for 15 h and at 4 °C during the night. The results represent measurements for at least 90 cells analyzed in at least two independent experiments. The analysis was performed by micromanipulation using a Nikon Eclipse E200 optical microscope with an attached micromanipulator.

### 2.12. Chronological Lifespan (CLS) Assays

Chronological lifespan of cells incubated in minimal medium (SDC) was measured as previously described [23]. Briefly, yeast was grown in SDC containing 0.67% Bacto-yeast nitrogen base (without amino acids) and 2% (*w*/*v*) glucose, supplemented with L-histidine (60 mg/L), L-leucine (180 mg/L) and uracil (60 mg/L). Chronological lifespan was monitored in SDC medium by measuring viability after 2, 4, 7, 14, 21 and 28 days of cultivation. For the quantitative measurement of survival, staining with propidium iodide was used. The data represent the mean values from at least three independent experiments.

### 2.13. Cell Viability

To determine cell death, propidium iodide was used for cell staining. Cells were suspended in PBS and stained with 5 µg/mL propidium iodide (Sigma-Aldrich, Burlington, MA, USA) for 15 min in the dark at room temperature as previously described. The heat-shocked cells were used here as a positive control (the cells from overnight growth culture were placed in 60 °C for 15 min). Photos of fluorescence signals were taken with Olympus BX-51 (Olympus, Tokyo, Japan) microscope equipped with a DP-72 digital camera and cellSens Dimension v1.0 software. Dead cells were visible in the red fluorescence channel (λex = 480 nm; λem = 520 nm). The data represent the mean values from three independent experiments.

### 2.14. *Raman Spectroscopy*

For the Raman spectroscopic measurements, a Nicolet NXR 9650 FT-Raman Spectrometer equipped with an Nd:YAG laser (1064 nm) and an InGaAs detector was used. Samples of yeast were lyophilized. FT-Raman spectra were measured at a laser power of 0.5 W in a range from 400 cm^−1^ to 2000 cm^−1^; the resulting spectra were averaged from 64 scans. The diameter of the used laser beam was 50 μm, and the spectral resolution was 8 cm^−1^. Raman spectra were processed by the Omnic/Thermo Scientific software and OriginLab programs.

To indicate the similarity between the heterozygous strains and wild-type yeast hierarchical clustering analysis (HCA), the Ward cluster method was used in the studied groups of chemical compounds. These methods were applied in the Raman ranges for polysaccharides, lipids, proteins, RNA, and nucleic acid. OriginLab software was used to perform the analysis and to draw the charts.

### 2.15. Statistical Analysis

The results represent the mean ± SD values for all cells tested in two independent experiments. The differences between the wild-type and isogenic heterozygous strains were estimated using one-way ANOVA and Dunnett’s post hoc tests. The values were considered significant when *p* < 0.05. The statistical analysis was performed using the Statistica 10.0 software, and statistical and multidimensional analysis was conducted using PAST 3.0 and Origin 2018 software (Raman spectroscopy).

## 3. Results

### 3.1. ORC/orcΔ Heterozygous Strains Grow Slowly Due to a Delay in G1/S Transition in the Cell Cycle

We compared the growth rate and the average doubling time of heterozygous strains, lacking one copy of genes encoding subunits of the ORC complex, on a rich medium with addition of 2% glucose. As seen in Figure 1A,B, almost all of the strains show a slower growth rate. Two approaches to estimating the growth rate are shown here. The first approach is based on changes in the optical density (OD_600_) in time (Figure 1A). The other approach showed changes in the density of cultures (number of cells per mL) during the cultivation period (Figure 1B). We showed results of both measurements because, from our previous experience, we learned that the growth curve obtained by OD_600_ measurements might be misleading. Aside from cell culture density, the results of OD_600_ measurements also depend on the morphology of the cells (their size, transparency, granularity, thickness of their cell wall, etc.). For example, the curves representing the *ORC1/orc1Δ* strain growth differ in Figure 1A,B. Our observations during the routine budding lifespan analysis did not match with the fast growth seen for this strain in Figure 1A but were in accordance with the growth curve visible in Figure 1B.

Almost all heterozygous *ORC/orcΔ* strains, except *ORC6/orc6Δ*, showed a statistically significant increase in the mean cell doubling time, measured during the budding lifespan analyses (Figure 1C). The doubling time of the *ORC6/orc6Δ* strain is similar to the wild-type strain. The ability of all strains to grow on a fermentable and non-fermentable carbon source was also confirmed (Figure 1D). Here, we also found that typically no more than two spores from each tetrad were viable, which implied that *ORC1-6* were essential genes (Figure 1E).

Since the ORC complex subunits are known to form a pre-replication complex during the G1 phase of the cell cycle, we expected that a lack of one copy of the *ORC* gene might affect the cell cycle. The DNA content analysis, performed using flow cytometry, showed cell cycle anomalies in all analyzed strains. These changes were mainly related to the accumulation of cells in the G1 phase of the cell cycle (Figure 2A,B).

Consequently, the number of cells in the G2 phase of the cycle was significantly reduced in all strains compared to the control (Figure 2A,B).

Since the G1 phase of the cell cycle was prolonged in all *ORC/orcΔ* strains, we posed a question on how the copy number of *ORC* genes influenced their expression.

As shown in Figure 3, reducing the levels of Orc1-6 proteins led to a significant reduction in their expression (*p* < 0.05). The expression level of all analyzed *ORC* genes was reduced by more than 50% on average. The most remarkable changes were observed in the *ORC3/orc3Δ* and *ORC5/orc5Δ* strains, where expression of the respective *ORC* gene was reduced by more than 70% compared to the WT.

### 3.2. The Metabolic Activity, Sporulation Efficiency and Response to Oxidative Stress Are Changed in ORC/orcΔ Strains

We performed a further analysis of the physiological parameters of the cells to indicate the possible influence of the diminished expression of *ORC* genes in cell phenotypes. First, we characterized the metabolic activity of *ORC/orcΔ* heterozygous strains using the FUN-1 dye, a two-color fluorescent viability probe. Cell metabolic activity was expressed as the change in the ratio of red (λ = 575 nm) to green (λ = 535 nm) fluorescence. The higher the red/green fluorescence ratio, the higher metabolic activity was shown by the cells tested. As shown in Figure 4A, all analyzed strains showed a slight increase in metabolic activity compared to the WT; however, this increase was statistically significant only for the *ORC4/orc4Δ* strain.

Sporulation is a key process necessary for yeast to survive in natural conditions. Therefore, we investigated the effect of reducing the levels of Orc1-6 proteins on the frequency of sporulation. We showed that the *ORC2/orc2Δ* and *ORC5/orc5Δ* strains had a lower sporulation frequency compared to the control strain (Figure 4B). The lack of a single copy of the gene in *ORC3* and *ORC6* also decreases the ability of yeast to sporulate, but this result is not statistically significant. Interestingly, for the *ORC5/orc5Δ* strain we saw an increase in the sporulation frequency. Therefore, it seems that ORC subunits, besides their role in replication, are involved in other biological processes, such as meiosis, or the cell’s response to adverse environmental conditions. This result suggests that ORC subunits belong to a class of the so-called moonlighting proteins.

The initiation of replication is an essential step in the life of proliferating cells. We previously showed the delay in the G1/S transition in the heterozygous *ORC/orcΔ* strains (Figure 2), which was likely due to a diminished access to ORC components. We asked whether decreased ORC components accessibility might lead to increased DNA damage when cells were exposed to oxidative stress. As a DNA damage marker, we used the Rad52-YFP fusion protein, which is known to be recruited to the site of damage. We checked the frequency of Rad52-YFP foci formation in *ORC/orcΔ* strains under normal conditions and after a one-hour treatment with oxidative stress inductor H_2_O_2_. Under control conditions (growth on YPD medium), we observed a decreased number of cells with Rad52-YFP foci in the *ORC2/orc2Δ* strain compared to the WT (Figure 4C). After oxidative stress, the number of cells with the Rad52-YFP foci was significantly increased in the *ORC1/orc1Δ* and *ORC2/orc2Δ* strains but reduced in the *ORC6/orc6Δ* strain compared to the WT (Figure 4C).

### 3.3. Strains with Reduced Expression of ORC Subunits Have Extended Budding and Chronological Lifespans

Aging analysis was crucial for this study. To achieve access to yeast aging phenotypes, we used two main approaches. The replicative aging model studies aging in mitotically active cells in higher eukaryotes, including humans. The standard approach is to count the number of daughter cells that a yeast mother cell is able to produce during its lifetime (budding lifespan). Since the yeast cell does not simply die after the completion of budding, we could determine not only the budding lifespan, but also in the cell’s lifespan during replication and lifespan after replication (time period from the last bud detachment to death), which sums up to the cell’s total lifespan. The experiments were performed using a single-cell analysis system. The second aging model used in this study, the chronological aging model, was applied to measure the aging of post-mitotic cells.

Our data clearly showed that *ORC/orcΔ* heterozygous strains delayed aging in both aging models. As shown in Figure 5A, all strains, except *ORC6/orc6Δ*, had significantly extended budding lifespans. In almost all cases, the mean budding lifespan exceeded 30 doublings performed by a single yeast mother cell, which was a statistically significant increase compared to BY4743 (*p* < 0.001). One of the consequences of the increase in the budding lifespan was an increase in the reproductive time. The additional increase in the doubling time (as was shown in Figure 1C) led to a statistically significant increase in the reproductive lifespan of the *ORC1/orc1Δ*, *ORC2/orc2Δ*, *ORC3/orc3Δ*, *ORC4/orc4Δ* and *ORC5/orc5Δ* strains compared to the WT (*p* < 0.001) (Figure 5B).

The *ORC6/orc6Δ* strain also had a slightly increased reproductive lifespan, but this change was not statistically significant. The cells do not die after producing their last bud. The last period of a cell’s life is defined as the post-reproductive lifespan. Its duration depends on many factors and often determines longevity expressed in time units. Here, we showed a significant decrease in the post-reproductive lifespan of *ORC1/orc1Δ* (*p* < 0.01) and the *ORC2/orc2Δ*, *ORC3/orc3Δ*, *ORC4/orc4Δ* and *ORC5/orc5Δ* (*p* < 0.001) strains (Figure 5C). Interestingly, as shown in Figure 5D, only one copy of the *ORC* genes had no effect on the total lifespan. As shown in Figure 5D, the heterozygous *ORC/orcΔ* strains’ total lifespan was similar to that of the WT control. Although the decreased expression of the *ORC* genes, whose products are involved in the initiation of replication, has a significant impact on several aging parameters and the doubling time, it has no effect on the lifetime of mitotically active cells, which is a novelty in aging research.

Figure 6A presents the data on the post-reproductive lifespan (PRLS) of all tested strains. A negative correlation between PRLS and the budding lifespan is evident. The trend line suggests a strong negative correlation between these parameters. The value of the Pearson correlation coefficient is −0.91. The study also showed a strong positive correlation between the reproductive lifespan and the budding lifespan. The value of the Pearson correlation coefficient is 0.93 (Figure 6B).

Chronological lifespan (CLS) is a measure of the survival of a yeast population during the stationary phase. For CLS analysis, cells were cultured using a synthetic medium supplemented with glucose and the necessary amino acids. Under these conditions, upon completion of the logarithmic growth phase, the carbon source in the medium was depleted, and cells ceased to grow. We asked whether the CLS of heterozygous *ORC/orcΔ* strains might be affected. We used two approaches to measure cells’ survival: propidium iodide dyeing and the ability to form a colony (expressed in CFU) (Figure 7A,B).

As shown in Figure 7, all the analyzed *ORC/orcΔ* heterozygous strains had significantly augmented survival up until day 7 compared to the BY4743 wild-type strain (*p* < 0.001). On day 14, although the survival of all analyzed strains was on average higher than the WT, statistical significance was demonstrated only for *ORC3/orc3Δ* and *ORC4/orc4Δ*. Despite the high survival rate at day 14, less than 20% of cells from all strains had the ability to form colonies. On day 21 of the experiment, all the *ORC/orcΔ* strains were dead, which meant that the WT strain lived longer (survival approx. 15%). Aside from the small percentage of living cells from the wild-type strain population, the cells did not have the ability to form colonies (Figure 7B). The results showed the importance of presenting CLS results using several methods, not just one.

The microscopic analysis showed that cells in the post-mitotic growth phase increased in size during the CLS experiment. Therefore, cell size and DNA content changes during chronological aging were examined. As shown in Figure 8 (right panel), cell size increased successively until the end of the experiment, indicating a strong correlation between cell size and survival. The smallest cells were observed at the start of the experiment (2 days) and the largest cells were observed on days 14 and 21. Interesting results were also provided by the analysis of the DNA content in cells (Figure 8, left panel). The results showed that, during chronological aging, probably due to starvation condition, a change in the DNA content occurred seen as ploidy reduction. This phenotype was observed in all of the studied strains, including wild type. It should be mentioned here that when performing DNA content analysis, we cannot distinguish between living and dead cells’ FACS profiles in the analyzed population. All cells are subject to permeabilization, which allows for an effective PI entrance into the cells and subsequent DNA intercalation. We observed both decreases in the viability of cells in the population and ploidy reduction over time. At the moment, we cannot conclude how these phenotypes are linked. Haploidization might be the cause of cell death, or it might be a way for cells to escape death or at least to prolong life for some time.

### 3.4. Reducing the Levels of Orc1-6 Proteins Influences the Chemical Composition of the Cells

It seems that the changes in the reproductive potential of heterozygous *ORC1-5/orc1-5**Δ* strains are associated with significantly reduced gene expression of the *ORC* genes and the resulting changes in the cell cycle pattern, subsequently reducing energy consumption and leading to an increase in the number of daughter cells produced. However, there is no clear evidence as to why, despite the reduced expression of *ORC6*, we did not observe a similar phenotype to that in the *ORC6/orc6**Δ* heterozygous strain. As already shown, certain changes in the budding lifespan correlate with altered cell chemistry composition [27,36]. Thus, differences in the chemical composition of the wild-type (WT) and *ORC/orc**Δ* heterozygous strains were shown by the analysis of the Raman spectra, which are presented in Figure 9.

For all investigated samples, the peaks in the Raman spectra correspond to vibrations of functional groups in proteins (1550–1610 cm^−1^) [37,38], lipids (1250 cm^−1^) [37,39], polysaccharides (487, 528, 1150 cm^−1^) [37,39], nucleic acids (660, 1092 cm^−1^) [39], and RNA (530, 720 cm^−1^) [37,40].

An HCA analysis made it possible to identify differences in the chemical composition of polysaccharides, lipids, proteins, RNA, and nucleic acid between the mutants and the WT yeast (Figure 10). In the case of the polysaccharide composition, only three yeast mutants (*ORC1/orc1Δ*, *ORC2/orc2Δ*, and *ORC3/orc3Δ*) were similar to the WT. Other mutants formed separate groups, for which the composition of polysaccharides was significantly different in comparison to the WT (Figure 10A). A comparison of the lipids composition helped distinguish three groups with a similar composition: (1) WT and *ORC6/orc6Δ*, (2) *ORC1/orc1Δ* and *ORC2/orc2Δ*, and (3) *ORC3/orc3Δ*, *ORC4/orc4Δ* and *ORC5/orc5Δ* (Figure 10B). The main similarity in the protein composition was between the WT and *ORC1/orc1Δ*, *ORC5/orc5Δ* and *ORC6/orc6Δ* strains, whereas the other mutants showed a significant difference in the protein composition compared to the WT. The *ORC4/orc4Δ* strain sample was the most different and formed a separate group (Figure 10C). The chemical composition of RNA from the *ORC6/orc6Δ* strain was similar to the WT. The rest of analyzed strains had a different RNA composition than the WT and formed two separate groups (Figure 10D). The WT shared similarities with the *ORC5/orc5Δ* strain in the nucleic acid composition. The *ORC4/orc4Δ* and *ORC6/orc6Δ* strains formed a second group with a similar composition of nucleic acids, which were different from the WT. The third group, which differed from the WT, was formed by the *ORC1/orc1Δ*, *ORC2/orc2Δ* and *ORC3/orc3Δ* strains (Figure 10E). Therefore, we believe that the lack of changes in the RNA spectrum and lipids may determine the similarity between the budding lifespan and the *ORC6/orc6**Δ* growth curve. Interestingly, strains bearing one copy of *ORC1-5* genes are resistant to calcofluor white (50 µg/mL), and *ORC6/orc6**Δ* sensitivity is similar to that of the wild type (data not shown). This observation suggests that ORC subunits may play an important role in cell wall remodeling, cell wall synthesis or the response to environmental stress.

## 4. Discussion

In general, DNA replication requires at least one origin and a protein complex in all three domains of life: archaea, bacteria and eukaryote. There is one origin of replication in prokaryotic cells [41], while in eukaryotes, especially higher eukaryotes, there may be tens of thousands of origins of replication [42]. In eukaryotes, protein factors bind to the replication origin, the most important of which is the origin recognition complex, to which other factors that initiate replication attach to establish a pre-replicative complex [8,41,42,43,44]. The ORC forms a heterohexamer consisting of the Orc1, Orc2, Orc3, Orc4, Orc5 and Orc6 proteins, forming stable binds to all of the origins of replication DNA [45]. The binding of the ORC to specific DNA sites is ATP-dependent. In the ORC, Orc1-5 proteins contain an AAA+ domain (ATPases associated with diverse cellular activities) and are required for binding to DNA. Interestingly, only Orc1, Orc2, Orc4, and Orc5 have direct contact with DNA [46,47], while Orc3 is a linker of the subunits that are in contact with DNA, which stabilizes the entire complex. In turn, the Orc6 subunit does not contain an AAA+ domain and does not bind to DNA, but it recruits Cdt1 and the replicative helicase core—the Mcm2-7 complex [48,49].

The molecular basis for the initiation of replication in eukaryotes seems to be fairly well-understood and reliably described. However, so far there have been no data on the effect of the reducing the level of essential proteins involved in the recognition of the replication origin on cell physiology and aging. There are strong similarities between senescence of mammalian cells and the budding yeast lifespan. Therefore, a good understanding of the mechanisms controlling the initiation of replication can be helpful in slowing aging and providing protection against cancer. Cellular senescence is a cell-cycle arrest that can be induced in cells in response to various internal and external factors, as well as developmental signals [50]. During oncogene induced senescence, oncogene activation causes hyperproliferation, which induces cellular senescence. Mitotic signals increase the usage of origins of replication resulting in the arrest of replication and accumulation of DNA damage, eventually activating the DNA damage response [51,52]. The ORC as a key player during the initiation of replication and the licensing factors, such as Cdc6 and Cdt1, cooperatively promotes the loading of the helicase core MCM complex (minichromosome-maintenance) [53,54]. During the S phase, the activated cyclin-dependent kinases and Dbf4-dependent kinase trigger the initiation of DNA replication. The phosphorylation of MCM by CDK and Dbf4 is necessary for origin firing. In turn, the ORC and Cdc6 and Cdt1 are downregulated by phosphorylation to prevent MCM recruiting [55]. Recent studies show an interesting link between the ORC and telomers [56]. In general, to achieve the efficient duplication of telomeric DNA regions, a complex protein network is necessary. It is suggested that, in mammalian cells, a crucial role in this process is played by TRF2, which is involved in both ORC and MCM loading at telomeres. The TRF2 directly binds to ORC through the Orc1 subunit; it is suggested that replication origins are assembled at telomeres through the TRF2–ORC interaction [57] and plays a key role in maintaining telomere stability through the formation of dormant telomeric origins [56]. However, the answer to the question of whether telomere replication is actually impaired due to the defects in ORC recruitment remains elusive. Our data clearly show that strains with low levels of Orc1-5 have significantly prolonged doubling times. This slowed growth corresponds to the cell-cycle delay in the G1/S transition, yet without influencing metabolism measured by the FUN-1 dye. Interestingly, no changes in growth kinetics in the *ORC6/orc6Δ* strain were observed despite the cell cycle anomalies. In addition, we found that heterozygous cells decreased the amount of *ORC* mRNA compared to the control, including *ORC6* mRNA. This suggests that the AAA+ domain, not cell cycle anomalies, may play a superior role in the observed phenotype, at least in the case of the yeast ORC. The ORC binds directly to DNA at replication origins throughout the cell cycle, and therefore plays a crucial role during eukaryotic DNA replication [4]. This also constitutes a foundation for the assembly of the pre-replicative complex during the G1 phase. A cell cycle analysis clearly showed that all of the heterozygous strains in question had an extended cell cycle G1 phase compared to the reference strain. Recently, we showed that some parameters of cell chemistry may correlate with each other and may account for the phenotype of mutants [27,36]. Molecular analyses have not provided clear answers to why the *ORC6* level, despite disturbances in the cell cycle and lower mRNA in the cell, does not have an altered budding lifespan and growth rate. This is why we used the Raman spectrometry here. The results of our analyses show that cells in the exponential phase, and thus comparable to those in the budding lifespan analysis, have similar RNA and lipids spectra. These results support previous findings suggesting that the biochemical status (metabolomic fingerprint) of cells may be associated, to some extent, with yeast longevity [58]. Our results also support previous data highlighting the significance of lipid metabolism and homeostasis in aging [36]. Therefore, it seems that, for the understanding of biochemical fundamentals of aging in yeast, pivotal roles must be assigned to lipid metabolism and lipid homeostasis [59]. The role of triacylglycerols was particularly evident in chronological aging. However, this relationship is less clear for the budding lifespan. To achieve a full budding lifespan, only TAG synthesis capability is needed [60].

In our extensive functional analysis of the ORC, we also focused on determining chronological lifespan (CLS). CLS is the length of time that non-budding yeast cells are able to survive. By providing a model for the aging of nondividing cells of higher organisms, the yeast CLS assay is suggested as a complement to the budding lifespan assay [61,62]. Despite the discovery of many metabolic pathways involved in CLS, our knowledge of replication initiation disorders of nonbudding cells is elusive. Theoretically, disturbances in replication initiation should not affect aging of yeast cells in the post-mitotic phase. It is worth noting that the CLS analysis begins with the entry of cells into the stationary phase. It has been proposed that, during CLS, two main pathways are activated: the Tor/S6K pathway [63] and the Ras/adenylate cyclase/PKA pathway [64]. Interestingly, all of the studied strains showed a longevity comparable to that of the WT. Here, we also confirm that both aging models, i.e., RLS and CLS, are not always similarly affected. Previous studies found that overexpression of the *SOD1* and *SOD2* genes decreased budding lifespan, but also slows down chronological aging [20]. In turn, other studies showed that increased respiration in the *sch9Δ* mutant was required for increasing CLS but not RLS [65]. It was recently shown that haploinsufficiency, i.e., a dominant phenotype caused by a heterozygous loss-of-function mutation, is rare. Interestingly, for yeast haploinsufficiency, phenotypes were observed for more than a half of the essential genes under optimal growth conditions. Moreover, 40% of the essential genes without noticeable phenotypes under optimal growth conditions exhibited haploinsufficiency under extreme growth conditions [66]. This may explain the absence of a direct *ORC6/orc6Δ* phenotype in the replication aging model and significant alterations in the cell response to cell stress conditions during CLS, for instance, starvation or acidification of the medium. Therefore, we strongly support the suggestion that single-cell phenotyping is a powerful approach, even in heterozygous conditions. In general, our data suggest that the ORC may have a prompting role in both budding and average chronological aging. During the CLS analysis, we observed cells with enlarged volumes appearing in the population, which confirms our previous data [22]. Our flow cytometry analyses confirmed the microscopic observations and clearly showed that, during CLS, the population of enlarged cells grows with age regardless of the strain. Furthermore, analysis of the DNA content of the cells showed ploidy changes during the chronological aging, explicitly, the ploidy shift toward the haploid. The phenomenon of ploidy loss was discovered for tetraploid strains of *Candida albicans* [67] but was shown for budding yeast [68,69]. Moreover, haploidization was also proved, among other processes, for strains lacking proteins involved in the initiation of replication, e.g., Ctf18 [69]. Changes in ploidy in yeast cells may be caused by several mechanisms such as condensation defects in the incompletely replicated or unrepaired damaged DNA, disturbances in the control of proper chromosome divisions, defects in the interaction between DNA microtubules, kinetochore and spindle, and disturbances in their organization, and may also result from external factors, such as radiation or heat shock [70]. The previous data show that ploidy reduction occurs in strains lacking LAS21, GGA2 or VPS1 genes, indicating that genomic changes, such as a ploidy shift, may be caused by a single-gene deletion [68]. Additionally, the long cultivation of diploid cells (about several hundred generations) could result in ploidy reduction [71,72]. In human cells, a ploidy shift is also observed, where a prolonged replication block, or DNA damage that causes mitotic catastrophe, leads to abnormal cell division, aneuploidization or polyploidization through the failure of cytokinesis or mitotic slippage [73].

To conclude, securing the ORC function by maintaining the proper Orc1-6 subunits level in yeast is crucial in aging. Our data clearly show that an increase in the budding lifespan is strongly correlated with a lowered expression of *ORC* genes. The ORC subunits (Orc1-Orc5), which contain the AAA+ domain, seem to be especially important to the aging phenotypes. The ORC proteins containing the AAA+ domain also have a significant influence on cell metabolism, including cell wall modeling. We hypothesize that the lack of an effect of Orc6 dysfunction on replicative aging is related either to the auxiliary function of this protein in the activity of the complex and/or lack of changes in the lipid shift metabolic trace and similarity to the WT. The reduced expression level of *ORC* genes has no effect on the lifetime of mitotically active cells but significantly delays average chronological aging. On this occasion, we were able to show, for the first time, that diploid cells reduce their genomes during chronological aging. This phenomenon might be related to the exposure of cells to stressful conditions arising with time, such as starvation or other endogenous and exogenous stresses, which accompany prolonged growth in an unfavorable environment (e.g., oxidative stress, proteotoxic stress, quorum sensing signals, etc.). At the moment, the mechanism of strain haploidization has not been revealed; however, we cannot exclude any scenario. There are several different working hypotheses. One hypothesis states that the appearance of haploid cells in the yeast culture during the chronological lifespan experiment is due to sporulation. However, several arguments stand against this hypothesis. Firstly, we did not notice tetrads during the microscopic analysis of strain culture samples during the chronological lifespan assay. Secondly, the spores should not germinate when the environment is poor and nutrients are inaccessible. The second hypothesis states that what we observed reflects DNA degradation in the dead cell fraction. However, the question remains of why such degradation is limited to a single set of chromosomes. The third hypothesis states that we observed the gradual loss of one copy of the genome. Song and Petes [71] described this phenomenon for the *rad52Δ/rad52Δ* strain. This strain, lacking the Rad52 recombinase required for homologous recombination, has a very high rate of chromosome loss; thus, all chromosomes are lost with some frequency, leading to haploid cell formation. We think that this could be the case because we see the gradual shift of DNA profiles toward the left part of the graph in the DNA content histograms, which reflects a diminishing DNA content (Figure 8). This change was visible for the cells after the 17th day of cultivation and escalated after 21 days. Moreover, we noticed that the average cell size had not changed much between the 7th and 21st days of cultivation. Thus, the cells do not resemble the typical haploid cell, at least not by their size (Figure 8). The fourth hypothesis states that during chronological aging, the genomic DNA undergoes condensation over time, which makes intercalation of the fluorescent dye difficult. Subsequently, DNA content analysis by flow cytometry will artificially show the ploidy shift. Whichever of the described hypotheses will turn out to be true, our research enabled us to show the existence of yet another mechanism contributing to cellular aging.

## Figures and Tables

**Figure 1 cells-11-01252-f001:**
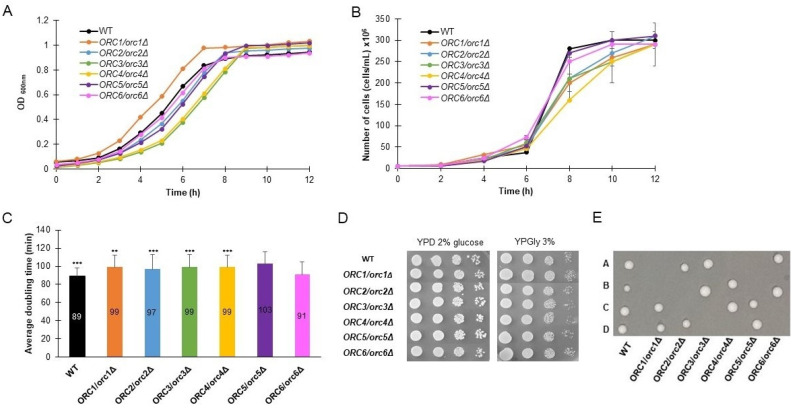
Growth phenotypes of heterozygous strains lacking one copy of genes encoding ORC complex subunits. Growth curves of heterozygous *ORC1/orc1Δ*, *ORC2/orc2Δ*, *ORC3/orc3Δ*, *ORC4/orc4Δ*, *ORC5/orc5Δ*, and *ORC6/orc6Δ* strains and the WT control (BY4743). The density of the culture was expressed by OD_600_ (**A**) or number of cells per mL (**B**). Average doubling time of the same strains, as in (**A**), during the reproducing period. Error bars represent standard deviations obtained from three independent experiments. Statistical significance was assessed using ANOVA and Dunnett’s post hoc test (** *p* < 0.01, *** *p *< 0.001). (**C**) Growth of the strains on solid YPD or YPGly media. Drops of serial diluted cell cultures (the same strains as in (**A**)) were placed on appropriate media, allowed to grow at 28 °C and photographed after 48 h (**D**). Growth of spores from tetrad dissection of BY4743 and *ORC/orcΔ* heterozygous diploids. Spores were dissected onto YPD plates and grown at 23 °C for 72 h (**E**).

**Figure 2 cells-11-01252-f002:**
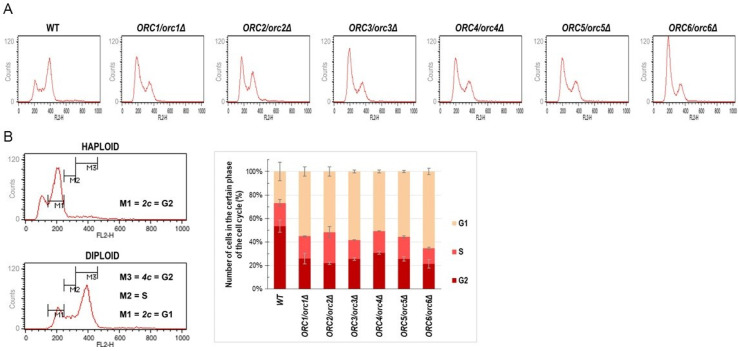
Cells lacking one copy of the *ORC* gene showed a delay in G1/S transition (**A**) Flow cytometry analysis of the diploid wild-type yeast strain BY4743 and the isogenic heterozygous strains *ORC1/orc1Δ*, *ORC2/orc2Δ*, *ORC3/orc3Δ*, *ORC4/orc4Δ*, *ORC5/orc5Δ*, and *ORC6/orc6Δ*. Quantitation of the cell cycle analysis. (**B**) On the left, gating conditions used to calculate the number of cells in the individual cell cycle phases. On the right, the result of quantification of three biological repetitions.

**Figure 3 cells-11-01252-f003:**
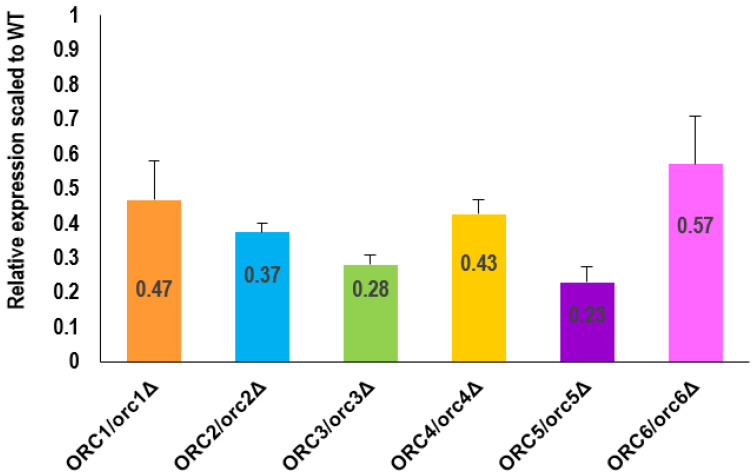
RT-PCR analysis showed a decrease in the *ORC* genes expression in the heterozygous *ORC/orcΔ* strains. The relative expression ratio between the WT and the *ORC1/orc1Δ*, *ORC2/orc2Δ*, *ORC3/orc3Δ*, *ORC4/orc4Δ*, *ORC5/orc5Δ*, and *ORC6/orc6Δ* heterozygous strains (normalized by *ACT1*), respectively, was quantified from five independent repetitions. Standard deviation was also counted. All results were statistically significant, with *p* < 0.05.

**Figure 4 cells-11-01252-f004:**
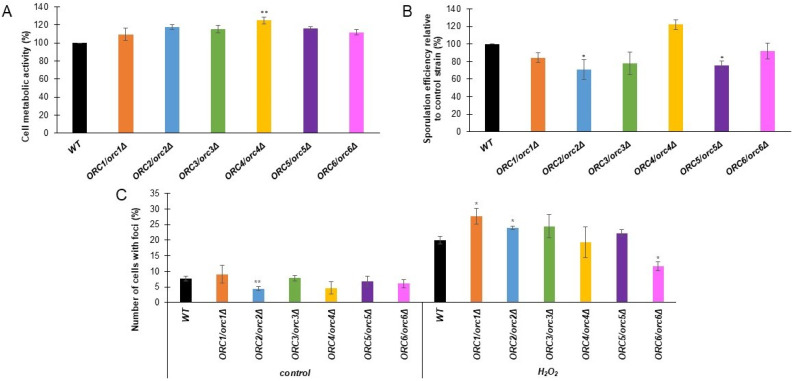
(**A**)—Metabolic activity of the cells (red/green fluorescence ratio) was estimated with FUN-1 stain for the diploid wild-type yeast strain BY4743 and the isogenic strains *ORC1/orc1Δ*, *ORC2/orc2Δ*, *ORC3/orc3Δ*, *ORC4/orc4Δ*, *ORC5/orc5Δ*, and *ORC6/orc6Δ*. Data are expressed as the ratio of red (λ = 575 nm) to green (λ = 535 nm) fluorescence and presented as mean ± SD from three independent experiments. Bars indicate SD. Statistical significance was assessed using ANOVA and Dunnett’s post hoc test (* *p* < 0.05; ** *p* < 0.01) compared to the control (wild-type strains) (**B**)—Sporulation frequency of the BY4743 control and the isogenic heterozygous strains *ORC1/orc1Δ*, *ORC2/orc2Δ*, *ORC3/orc3Δ*, *ORC4/orc4Δ*, *ORC5/orc5Δ*, and *ORC6/orc6Δ*. Error bars represent standard deviations obtained from three independent experiments. Statistical significance was assessed using ANOVA and Dunnett’s post hoc test (* *p* < 0.05, ** *p* < 0.01). (**C**)—Quantification of Rad52 foci frequency. The control strain BY4743 and the isogenic strains *ORC1/orc1Δ*, *ORC2/orc2Δ*, *ORC3/orc3Δ*, *ORC4/orc4Δ*, *ORC5/orc5Δ*, and *ORC6/orc6Δ* were inspected for Rad52 foci formation in at least 900 cells per each strain. The number of foci observed before and after H_2_O_2_ treatment was counted, and the average number of foci per cell was calculated. Error bars represent standard deviations obtained from three independent experiments. Statistical significance was assessed using ANOVA and Dunnett’s post hoc test (* *p* < 0.05, ** *p* < 0.01).

**Figure 5 cells-11-01252-f005:**
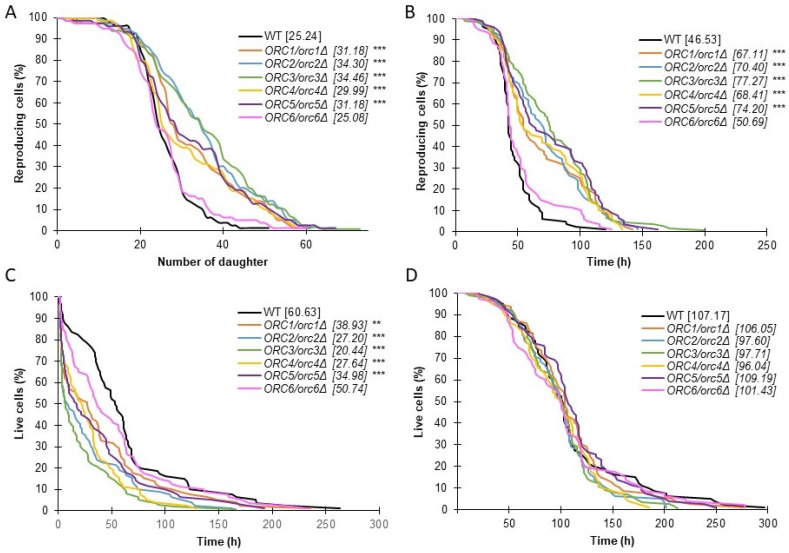
Aging phenotypes of the *ORC/orcΔ* heterozygous strains. Comparison of the budding lifespan (**A**), reproductive lifespan (**B**), post-reproductive lifespan (**C**) and total lifespan (**D**), of the diploid wild-type yeast strain BY4743 and the isogenic heterozygous strains *ORC1/orc1Δ*, *ORC2/orc2Δ*, *ORC3/orc3Δ*, *ORC4/orc4Δ*, *ORC5/orc5Δ* and *ORC6/orc6Δ*. Statistical significances were assessed using ANOVA and Dunnett’s post hoc test (** *p* < 0.01, *** *p* < 0.001). The mean value for total 80 cells from two independent experiments is shown in parentheses.

**Figure 6 cells-11-01252-f006:**
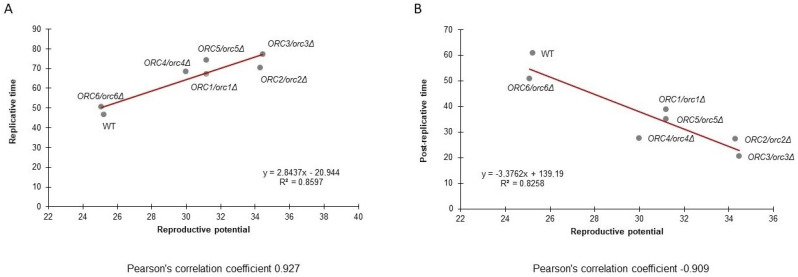
Pearson’s correlation coefficient between the budding lifespan and reproductive lifespan (**A**) and between the budding lifespan and post-reproductive lifespan (**B**) of the wild-type strain (BY4743) and the isogenic strains *ORC1/orc1Δ*, *ORC2/orc2Δ*, *ORC3/orc3Δ*, *ORC4/orc4Δ*, *ORC5/orc5Δ*, and *ORC6/orc6Δ*.

**Figure 7 cells-11-01252-f007:**
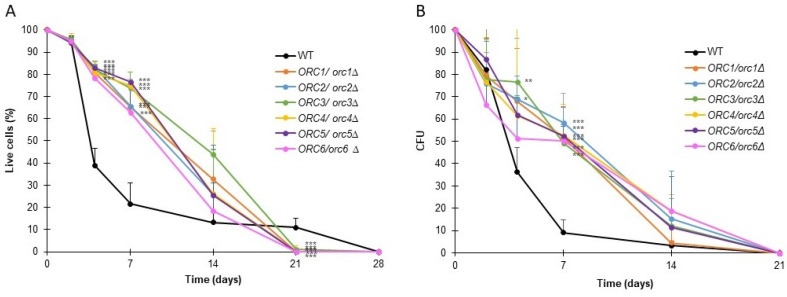
The chronological lifespan of the heterozygous *ORC/orcΔ* strains was extended. Chronological lifespan of the diploid wild-type yeast strain BY4743 and the isogenic strains *ORC1/orc1Δ*, *ORC2/orc2Δ*, *ORC3/orc3Δ*, *ORC4/orc4Δ*, *ORC5/orc5Δ*, and *ORC6/orc6Δ*. Survival was determined by propidium iodide staining (**A**). Clonogenicity was determined by colony-forming unit (CFU) (**B**). Error bars represent standard deviations obtained from three independent experiments. Statistical significance was assessed using ANOVA and Dunnett’s post hoc test (* *p* < 0.05, ** *p* < 0.01, *** *p* < 0.001).

**Figure 8 cells-11-01252-f008:**
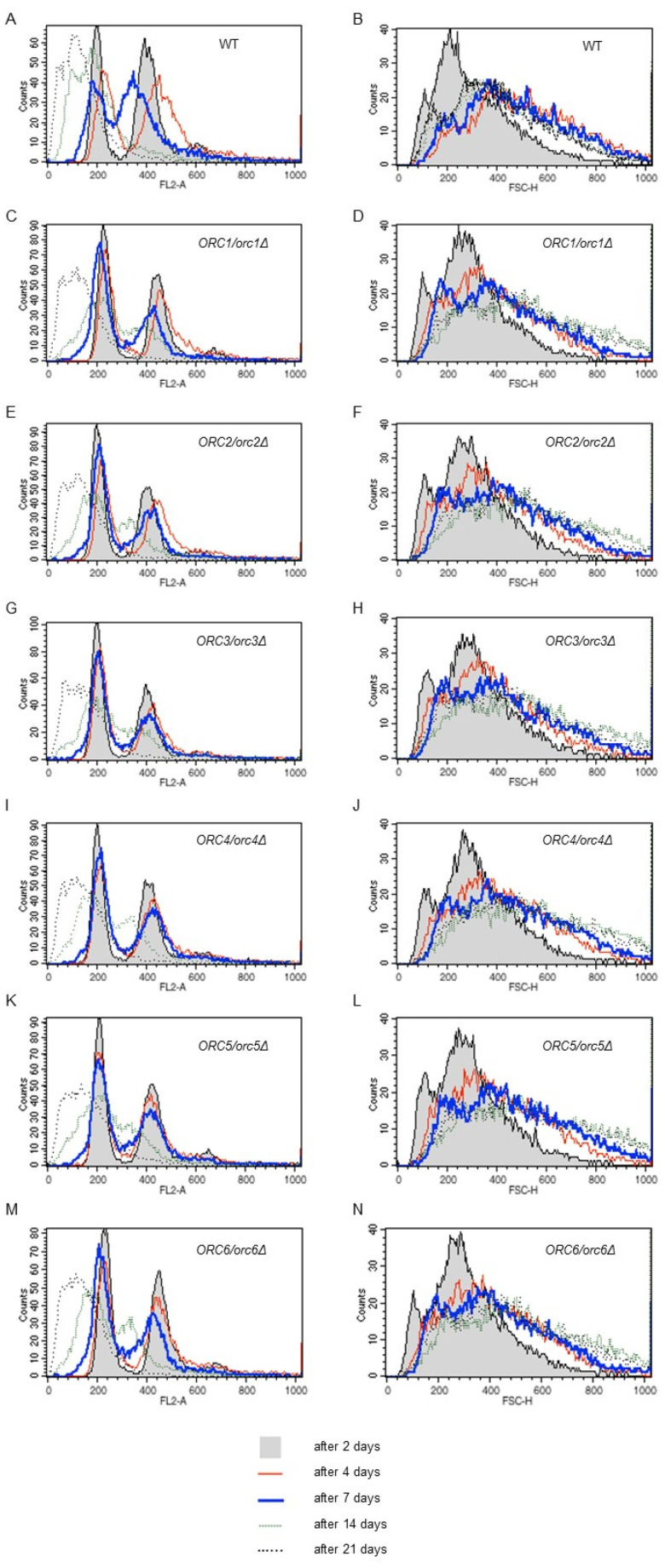
Ploidy reduction and cell size increase accompanies chronological aging. Results of the DNA content analysis via flow cytometry during the chronological lifespan analysis at selected time points (**A**,**C**,**E**,**G**,**I**,**K**,**M**). For the DNA content analysis yeast cells were stained with propidium iodide. The representative histograms are shown. Three independent experiments were performed. Cell size of the diploid BY4743 strain and the isogenic strains *ORC1/orc1Δ*, *ORC2/orc2Δ*, *ORC3/orc3Δ*, *ORC4/orc4Δ*, *ORC5/orc5Δ*, and *ORC6/orc6Δ* during the chronological lifespan analysis in selected time points (**B**,**D**,**F**,**H**,**J**,**L**,**N**). Cell size as measured by forward scatter (FSC histogram reflects the cells size in the population). The cells were analyzed via flow cytometry as described in the Materials and Methods section. As many as 10,000 cells per sample were assayed.

**Figure 9 cells-11-01252-f009:**
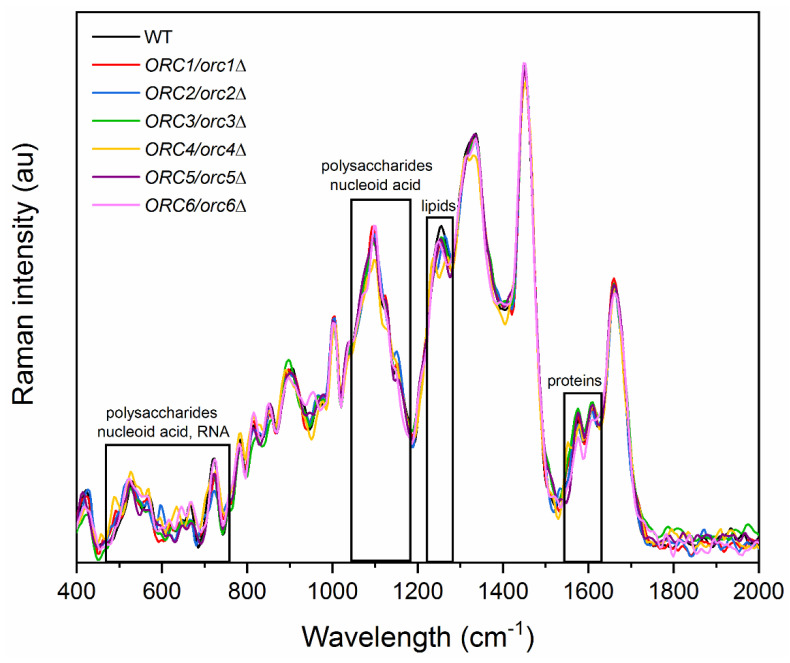
Raman spectra of the wild-type (WT) and heterozygous strains *ORC/orcΔ* 1–6 of yeast with the regions corresponding to vibrations of functional groups for RNA, nucleic acid, polysaccharides, proteins, and lipids.

**Figure 10 cells-11-01252-f010:**
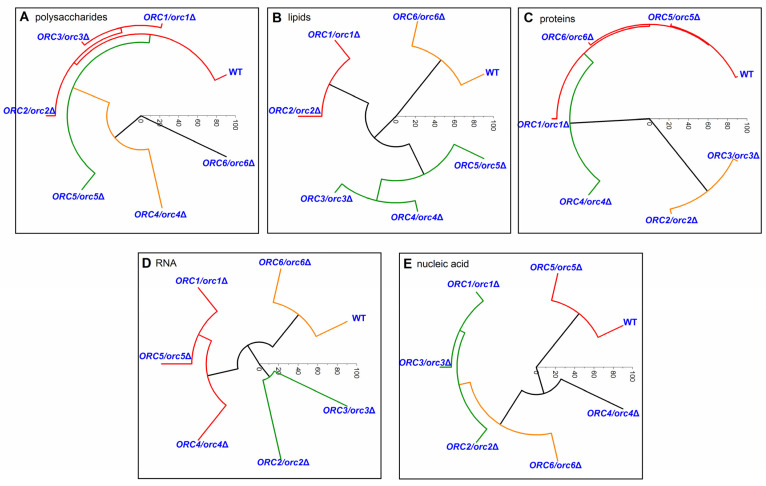
Hierarchical clustering analysis (HCA) of the control strain (BY4743) and the heterozygous strains *ORC1/orc1Δ*, *ORC2/orc2Δ*, *ORC3/orc3Δ*, *ORC4/orc4Δ*, *ORC5/orc5Δ*, and *ORC6/orc6Δ* obtained from Raman spectroscopy of different spectral functional groups: (**A**)—polysaccharides; (**B**)—lipids; (**C**)—proteins; (**D**)—RNA; (**E**)—nucleic acid.

**Table 1 cells-11-01252-t001:** Strains used in this study.

Strain	Genotype	Source
BY4743 a/*α*	Mat a/*α; his3Δ1/his3Δ1; leu2Δ0/leu2Δ0; lys2Δ0/LYS2; MET15/met15Δ0; ura3Δ0/ura3Δ0*	Open Biosystems
*ORC1/orc1Δ*	Mat a/*α; his3Δ1/his3Δ1; leu2Δ0/leu2Δ0; lys2Δ0/LYS2; MET15/met15Δ0; ura3Δ0/ura3Δ0; yml065w::kanMX4/YML065w*	Open Biosystems
*ORC2/orc2Δ*	Mat a/*α; his3Δ1/his3Δ1; leu2Δ0/leu2Δ0; lys2Δ0/LYS2; MET15/met15Δ0; ura3Δ0/ura3Δ0; ybr060c::kanMX4/YBR060c*	Open Biosystems
*ORC3/orc3Δ*	Mat a/*α; his3Δ1/his3Δ1; leu2Δ0/leu2Δ0; lys2Δ0/LYS2; MET15/met15Δ0; ura3Δ0/ura3Δ0; yll004w::kanMX4/YLL004w*	Open Biosystems
*ORC4/orc4Δ*	Mat a/*α; his3Δ1/his3Δ1; leu2Δ0/leu2Δ0; lys2Δ0/LYS2; MET15/met15Δ0; ura3Δ0/ura3Δ0; ypr162c::kanMX4/YPR162c*	Open Biosystems
*ORC5/orc5Δ*	Mat a/*α; his3Δ1/his3Δ1; leu2Δ0/leu2Δ0; lys2Δ0/LYS2; MET15/met15Δ0; ura3Δ0/ura3Δ0; ynl261w::kanMX4/YNL261w*	Open Biosystems
*ORC6/orc6Δ*	Mat a/*α; his3Δ1/his3Δ1; leu2Δ0/leu2Δ0; lys2Δ0/LYS2; MET15/met15Δ0; ura3Δ0/ura3Δ0; yhr118c::kanMX4/YHR118c*	Open Biosystems

**Table 2 cells-11-01252-t002:** Primers used for RT-qPCR gene expression analysis.

Gene Name	Forward Primer 5′->3′	Reverse Primer 5′->3′	Source
*ACT1*	GTAAGGAATTATACGGTAACATC	TAGATGGACCACTTTCGTCG	*in this study*
*ORC1*	TGGGTATACGCACGAAGAGC	TCCTCACGTCTTCAGGCAAC	*in this study*
*ORC2*	ATTTACGCTCCGCTCCTCTG	CTTCAGCACCACTGCTGGTA	*in this study*
*ORC3*	ACTGAGCAGATGTCCTACATTCA	GCCCGTTAATCGGGTTCTCT	*in this study*
*ORC4*	AGCTCGTCTATCACCGCAAG	CCAGGGTCGCTGTCTTTACA	*in this study*
*ORC5*	GGATTCCTCACGAAGTGCAGA	GGTAGAGCTGCTTATGGACG	*in this study*
*ORC6*	AACCAGGAAACGACGGTTTG	TTGTTTCGTTCTCCCGCTTC	*in this study*

## Data Availability

Not applicable.

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
