# Peer review of "Depletion of the Origin Recognition Complex Subunits Delays Aging in Budding Yeast"

_cells, 2022, doi:10.3390/cells11081252_

Round 1

Reviewer 1 Report

Major comments:

This is a timely paper discussing the several phenotypes associated with lowering the copy number of ORC genes in yeast (BY4743 strain). As a positive, the study evidence is convincing, clear, and well-designed experiments support individual experiments including budding and chronological lifespan.

However, as the main focus of the manuscript (Depletion of the origin recognition complex subunits delays aging in the budding yeast), identified individual phenotypes such as ORC’s slow growth, increase doubling time, low expression, metabolic activity, oxidative stress not correlated with their budding and chronological lifespan.

It would be important to add to this manuscript to include the mechanism that why lowering the copy number increases the budding and chronological lifespan, before publication. Moreover, it will be most important to correlate the observed phenotypes by authors for ORC’s heterozygous to budding and chronological lifespan to keep this manuscript in this special section (Yeast as a Model in Aging Research).

It is also important and interesting to include the phenotype associated with increasing the expression of copy numbers as the author claims that decreasing the expression of ORC genes delays aging.

Minor comments:

Confirm each strain for its heterozygous deletion.

Why were two different media were used (Determination of the total lifespan in YPD medium) and (Chronological lifespan assays in SDC medium).

Perform total lifespan in SDC medium.

Include the composition of SDC medium including all amino acids information

What is the positive control of PI staining? If dead cells then what was the preparation protocol?

5 mg/mL propidium iodide (please check 5mg/ml or 5microgram/ml)

PI staining is not sufficient to determine the total dead cells as cells can die by both necrosis (allow PI entering in cells due to plasma membrane damage) and apoptosis (cells death without plasma membrane damage). Perform viability using PI and annexin assays which are more appropriate for viability testing (PI=necrotic cells, Annexin= apoptotic cells).

Lifespan or Life span? (both are acceptable but should be consistent in manuscript)

Author Response

Dear Professor

Re: Manuscript id: cells-1643187 

Please find attached a revised version of our manuscript entitled "Depletion of the origin recognition complex subunits delays aging in the budding yeast", which we would like to resubmit for publication as a research paper in Cells.

We appreciate all the comments on the manuscript. According to suggestion, the manuscript has been edited by an English native speaker.

We are also grateful for constructive comments. We generally agree with the Reviewers' comments and list the point-to-point replies for easy reference. The manuscript has been greatly improved following the Reviewers' suggestions. We hope that you will find it worth publication in the Cells at the present shape.

We are looking forward to your decision.

Yours sincerely,
Mateusz Mołoń

Adrianna Skoneczna

Reviewer 1

  1. It would be important to add to this manuscript to include the mechanism that why lowering the copy number increases the budding and chronological lifespan, before publication. Moreover, it will be most important to correlate the observed phenotypes by authors for ORC’s heterozygous to budding and chronological lifespan to keep this manuscript in this special section (Yeast as a Model in Aging Research).

It should be emphasized that this is the first work that shows the relationship between depletion of replication initiation at the level of ORC proteins and aging in yeast. So far, the exact mechanism of ORC implication on aging is rather elusive. We assume, as mentioned in the manuscript, that there is a strong correlation between budding lifespan and ORC proteins with AAA+ domain. In addition, our subsequent research on, incl. analyzes of the MCM2-7 complex (each of the MCM proteins belongs to the broader AAA + family, i.e. ATP-az) (unpublished data) confirm these observations. Therefore, in the future, we will present the probable mechanism of action of proteins involved in the initiation of replication.

  1. It is also important and interesting to include the phenotype associated with increasing the expression of copy numbers as the author claims that decreasing the expression of ORC genes delays aging.

Yes, indeed. The Reviewer rightly believes that protein overexpression may have an impact on aging. Unfortunately, the time-consuming yeast aging analyzes prevent us from providing these data here. Unfortunately, it is difficult to assume what the effect of increasing the number of ORC proteins in the cell would be. In the future, it is definitely worth checking it out.

3.Minor comments: Confirm each strain for its heterozygous deletion.

Thank you for this suggestion. We supplemented figure 1E with the tetrad analysis.

  1. Why were two different media were used (Determination of the total lifespan in YPD medium) and (Chronological lifespan assays in SDC medium).
    Perform total lifespan in SDC medium.

Thank you for this question. We have used the methodology, which is the standard approach to define both Total and Chronological aging. During determination chronological lifespan, the SDC medium is used, allowing for limitation of cell budding by starvation and keeping cells in the post-mitotic phase. In contrast, when determining the total lifespan (total lifespan represents the lifespan of mitotically active cells), we want to allow cell budding; hence a rich medium is used. Especially for the Reviewer's knowledge, our unpublished data showed the same tendencies in haploid mutants cultured in SDC and YPD media. However, accordance with good practice for aging research, we present data obtained in the medium suggested for particular aging tests.

  1. Include the composition of SDC medium including all amino acids information

In response to the Reviewer's request, we have added the necessary information in section 2.11.

Chronological lifespan of cells incubated in minimal medium (SDC) was measured as described previously [23]. Briefly, yeast were grown in SDC containing 0.67% Bacto-yeast nitrogen base (without amino acids), 2% (w/v) glucose, supplemented with L-histidine (60 mg/L), L-leucine (180 mg/L) and uracil (60 mg/L). Chronological lifespan was monitored in SDC medium by measuring viability after 2, 4, 7, 14, 21 and 28 days of cultivation. For the quantitative measurement of survival, staining with propidium iodide was used. The data represent the mean values from at least three independent experiments.

  1. 5 mg/mL propidium iodide (please check 5mg/ml or 5microgram/ml).
    What is the positive control of PI staining? If dead cells then what was the preparation protocol?

Sorry for the mistake. It should be 5 micrograms/mL.

We used heat shocked cells as positive control. We have added the necessary information in section 2.12.

  1. PI staining is not sufficient to determine the total dead cells as cells can die by both necrosis (allow PI entering in cells due to plasma membrane damage) and apoptosis (cells death without plasma membrane damage). Perform viability using PI and annexin assays which are more appropriate for viability testing (PI=necrotic cells, Annexin= apoptotic cells).

Regarding the use of PI, we agree with the Reviewer that the PI staining is not an ideal way to indicate all dying cells. Cells in the early apoptosis stage will not be stained with PI; however, these at the late apoptosis stage will be stained. But, our goal was to show which cells were already dead, not differentiate between cell death pathways engaged in the dying process. At the present stage, we can not say whether cells were dying via apoptosis, necrosis, ferroptosis, mitotic catastrophe, or due to any other way of the death execution. In our research, we concentrated on lifespan, not on the mechanism of death (which is also fascinating/intriguing). Additionally, the apoptosis test is tricky when yeast cells are involved, as they possess a cell wall, which has to be removed to allow Annexin V staining. Stripping cell wall from cells affects apoptosis test results. In fact, we are still struggling to set up the apoptosis test for yeast cells in a way that will provide us the reproducible results. Nevertheless, figure 7 shows, besides the results of PI staining, the results of experiments testing the ability of cells to form a colony (CFU). The later test is a standard gold method used in many labs to show the viability of the cells. From our point of view, this test shows only the clonogenicity, not the survival. That is why in our approach, we added a PI staining test. We believe some cells in the analyzed population were still alive even though they could not divide anymore. We think the results of PI staining are informative; however, if the Reviewer insists, we are ready to remove Figure 7A  from the manuscript.

  1. Lifespan or Life span? (both are acceptable but should be consistent in manuscript)

We have unified the entire text to the term 'lifespan'.

Reviewer 2 Report

Please, see the attached file with comments and suggestions for authors.

Author Response

Dear Professor

Re: Manuscript id: cells-1643187 

Please find attached a revised version of our manuscript entitled "Depletion of the origin recognition complex subunits delays aging in the budding yeast", which we would like to resubmit for publication as a research paper in Cells.

We appreciate all the comments on the manuscript. According to suggestion, the manuscript has been edited by an English native speaker.

We are also grateful for constructive comments. We generally agree with the Reviewers' comments and list the point-to-point replies for easy reference. The manuscript has been greatly improved following the Reviewers' suggestions. We hope that you will find it worth publication in the Cells at the present shape.

We are looking forward to your decision.

Yours sincerely,
Mateusz Mołoń
Adrianna Skoneczna

Reviewer 2

  1. The main concern about the manuscript results is the interpretation of different phenotypes that are seen for some of the heterozygous ORC1-6/orc1-6 mutants but not consistently for all of them. For example, results in Figures 4A, 4B, 4C and 1B-C (see concern 3) with phenotypic differences for some mutants but not for others of the Orc1-6 complex. The complex ORC1-6 is supposed to be formed with one copy of each component, so reduction in any of the proteins is expected to have an impact in the amount of formed complex. Otherwise, other different functions may be assigned to some of the Orc1-6 proteins. To distinguish between these two possibilities, it should be checked the reduction in DNA bound ORC1-6 complex in each of the heterozygous ORC1-6/orc1-6. If same reduction in the complex is seen for all of them, then a particular additional function may be assigned to one Orc protein when there is a specific phenotype in the corresponding ORC/orc heterozygous.

 We agree with the Reviewer that the proposed experiment would distinguish between these two possibilities and bring new information concerning the ORC complex. However, the proposed biochemical approach could be itself the full-time project that extends the research area far beyond what we attempt to include in this very manuscript. According to the RT-PCR analysis results, lacking a single ORC gene led to a diminished mRNA level for this gene. Interestingly, the expression ratio was different for various ORCs compared to the mRNA level for these genes present in the control cells. The expression ratio varied from about ¼ to 3/5 and was lowest for ORC5 and ORC3 and highest for ORC6. This result showed the regulation of transcription is not precisely the same for all ORC genes and suggested the diverse cellular demand for various ORCs that may be linked to their additional cellular functions.

  1. In the abstract and discussion, it is stressed that heterozygous ORC/orc diploid cells undergo haploidization during chronological aging. This change in DNA content (Figure 8) also happens in wt diploid cells, therefore, it should be changed this sentence in the abstract and elsewhere to not mislead the audience in the interpretation of results. In other words, haploidization is not specific of low doses of ORC1-6 components, this change in DNA content also occurs in wt cells.

We appreciate the comment. Sorry for our overlooked mistake. Indeed, the haploidization concerns all strains and is not specific to low doses of Orc1-6 subunits. This phenomenon seems to be linked to starvation. Therefore, we conclude that this ploidy shift could be related to nutrient starvation or the inability to survive under stress conditions. All made changes were marked in red.

  1. In Figure 1B the curves of number cells vs. time corresponding to wt and ORC5/orc5 are very similar, and also to the curve corresponding to ORC6/orc6. By the contrary, curves of diploids with deletion ORC1-4 are similar and different to the first ones.

Here, we have shown the growth rate rather informally. More important in our opinion are the doubling time values shown in Figure 1C. The growth curve is only 12 h of optical density change, which does not fully reflect the dynamics of cell doubling (it is influenced by many factors, including cell size, morphology etc.). If necessary, we can leave only the doubling time graph (Fig. 1C) and omit the growth curves from this report (Fig. 1A and 1B).

However, in Figure 1C, the average doubling time is similar between wt and ORC6/orc6 but for ORC5/orc5 is the most different with respect to wt. Is there any error in the estimation of doubling times for ORC5/orc5??

No, ORC5/orc5 actually has an increased doubling time, and wt and ORC6/orc6 have a similar one (this similarity is also seen in budding lifespan).

This is relevant because in the Discussion section, lines 537-541, it is said that reduction in AAA+ domain ORC1-5 components (see also Minor concern 3) significantly cause growth disturbances and longer doubling times.

Yes, indeed. This statement is misleading. Therefore, we changed this sentence:

Our data has clearly shown that the strains with low levels of Orc1-5 have significant prolonged doubling times.

  1. In the Discussion section, lines 538-539 it is said that in the heterozygous strains for AAA+ domain ORC1-5 components (see also Minor concern 3) the “slowed growth strongly corresponds to the cell cycle delay in G1/S transitions”. But next it is said that for the strain ORC6/orc6 with no changes in growth kinetics there is similar delay in G1/S transitions. Therefore, I do not think that it can be said the first sentence, that slow growth “strongly corresponds” to delay in G1/s because it does not correlate with results for ORC6/orc6. This sentence should be changed because it is misleading for the audience.

Thank you for this suggestion. For better clarity we changed this sentence to:

This slowed growth corresponds to the cell cycle delay in G1/S transition, yet without influencing metabolism measured by the FUN-1 dye. Interestingly, no changes in growth kinetics in the ORC6/orc6D strain were observed despite the cell cycle anomalies.

  1. In relation with concern in (1), I wonder whether the measurement of DNA content by flow cytometry with propidium iodide is reflecting what happens in alive cells only, because I understand that in these assays the DNA content of all cells is measured, including the percentage of dead cells which are still detected as “cell” by the flow cytometry. An explanation about this will help to clarify the results.

The measurement of DNA content by flow cytometry reflects the fate of all cells, living and dead, in the population. We clarified the text in the manuscript:

“It should be mentioned here that when performing DNA content analysis, we can not distinguish between living and dead cells' FACS profiles in the analyzed population. All cells are subject to permeabilization, which allows the effective PI entrance to the cell and subsequent DNA intercalation. We observed both decreases in viability of cells in population and ploidy reduction with time passing. At the moment, we cannot conclude how these phenotypes are linked? Haploidization might be the cause of death, or it might be the way to escape from death or at least to prolong life for some time.”

  1. In some parts of the manuscript, for example in the Discussion section lines 512-513 it is interpreted the use of heterozygous diploids for ORC1-6 genes as “effect of the copy number”. This expression is misleading, the genes ORC1-6 are in single copy in haploid cells, no changes in gene copy number are described as far as I know. The heterozygotes ORC1-6/orc1-6 are used to reduce the levels of Orc1-6 essential proteins. Therefore, there is not an “effect of copy number” as it will be discussed for genes with variable gene copy number (for example for rRNA genes). It should always be used the expression of “reducing the levels of Orc1-6 proteins”.

Thank you for this suggestion. Indeed, the use of the expression „reducing the levels of Orc1-6 proteins' is more accurate. This statement has been changed throughout the text.

  1. An interpretation of the results in Figure 3 should be given. It would be expected that mRNA levels of ORC1-6 in the heterozygous ORC1-6/orc1-6 would be aprox. 50% the 3 mRNA levels in wt strain. However, the reduction in mRNAs for ORC2-5 is much lower, especially for ORC5 and ORC3. In fact, expression of ORC6 in the corresponding heterozygous is 57% the wt, and the higher of ORC genes in the diploids heterozygous.

May this be correlated with other phenotypic differences seen stronger for orc1-5 than for orc6??

The authors are grateful for this suggestion. The analysis in Figure 3 shows the relative change in mRNA level compared to WT. Indeed, the mRNA level in the case of ORC3 and ORC5 is the lowest. Unfortunately, these changes do not correlate significantly with other data. Therefore, we suggest that general Orc protein limitation (and AAA + domain-related functions) rather than dose is critical in maintaining longevity. Therefore, it seems that further studies using overexpression of Orc1-6 proteins (in a heterozygous ORC/orcD set) could respond to whether the changes in the level of these proteins will affect aging or other phenotypes. Here we hypothesize that the lack of effect of Orc6 dysfunction on replicative aging is related either to the auxiliary function of this protein in the activity of the complex and/or the lack of changes in the lipid shift metabolic trace and similarity to WT. We also strengthened the discussion in the context of haploinsufficiency phenotypes

  1. About results in Figure 7. The two ways of measurement of alive cells work consistently except for the wt at time 21 days, where propidium iodide staining shows more 10% alive cells, but CFU data indicates that cells are dead. May this be an error? If not, how is this interpreted? Moreover, in the abstract and elsewhere it is said that heterozygotes ORC1-6/orc1-6 show longer chronological lifespan than wt, but if data in Figure 7A is considered, then, the opposite result is true.

No, this is not an error. CFU, used as the gold standard in expressing survival in yeast, in fact, only shows the cell's ability to budding. Therefore, propidium iodide staining is the better (though probably not the best) method of determining survival.

Reviewer's suggestions are correct, so we change and use as average chronological aging/ average chronological lifespan. In this context, the data is not misleading.

  1. With respect the sentence in line 489: “We therefore believe that the lack of changes in the RNA spectrum and lipids determines the similarity of the budding lifespan and the ORC6/orc6Δ growth curve.” I think that results of Raman spectra are not exhaustive in describing the specific changes in RNA or lipids, so this conclusion is too strong (“determines”) for the results available.

Yes, indeed, but these data correlate with the aging analyzes. We changed this sentence as suggested by the Reviewer.

We therefore believe that the lack of changes in the RNA spectrum and lipids may determines the similarity of the budding lifespan and the ORC6/orc6Δ growth curve

Minor concerns:

1.- Line 187 is said: “analysis was being carried out in accordance with the principles of good practice in RNA handling”. It is needed a reference where these principles can be read.

The general sentence was removed because it adds nothing to the methodology.

Nevertheless, we provide key references for the Reviewer

RNA: A Laboratory Manual By Donald C. Rio, University of California, Berkeley; Manuel Ares, Jr., University of California, Santa Cruz; Gregory J. Hannon, Cold Spring Harbor Laboratory; Timothy W. Nilsen, Case Western Reserve University School of Medicine

https://www.metabion.com/assets/Downloads/Datasheets/RNA/Guidelines-for-Handling-RNA.pdf

https://www.qiagen.com/us/resources/download.aspx?id=d86e4457-e017-4f4a-84f9-8f5e2e2297e0&lang=en

https://www.biotech.cornell.edu/sites/default/files/2020-06/Zeiss%20RNA%20handling.pdf

2.- Line 307: “ORC gens” it should be said ORC genes.

Sorry for misspelling.

3.- Lines 537 says: “Our data has clearly shown that the strains with the AAA+ domain (Orc1-Orc5) have significant growth disturbances…”. It is referred to the heterozygotes, right? then it should be said: “strains with low levels of Orc1-5” or “heterozygotes ORC1-5/orc1-5”.

We changed this sentence:

Our data has clearly shown that the strains with low levels of Orc1-5 have significant prolonged doubling times

4.- In the Abstract, lines 26-27 is said: “Here, we also show that the limitation of the copy number of the ORC genes significantly extends both budding and chronological lifespans”, but in Discussion, line 575-576 it is said: “Interestingly, all of the studied strains showed a longevity comparable to WT.” Are these two sentences compatible in this yeast context?

Yes, this two sentences are compatible in this yeast context.

Discussion, line 575-576 it is said: “Interestingly, all of the studied strains showed a longevity comparable to WT.”

This sentence is related to a chronological lifespan. According to a standard approach, we can use the term longevity here.

Reviewer 3 Report

The manuscript provides an extensive phenotypic analysis of yeast strains heterozygous for ORC1-6, genes encoding origin recognition complex proteins. Interesting results about changes in two yeast specific models of aging, chronological lifespan (CLS) and replicative lifespan (RLS) were obtained.

Points to consider for improvement/clarification:

A potential ploidy loss of diploid yeast strains during chronological aging was observed and it is assumed that this effect is not the result of sporulation, which could be induced due to prolonged starvation and would give rise to haploid progeny. A sporulation defective strain could be used to clarify whether sporulation  is involved in ploidy loss during chronological aging. This could either be done experimentally or the possibility mentioned in the discussion/outlook.

Conclusions about ploidy reduction during chronological aging are based on flow cytometry detecting DNA fluorescence. To exclude the possibility that DNA degradation in the dead cell fraction has caused a down-shift in fluorescence intensity, it would be informative to assess chromosomal integrity, for example by PFGE. After 14 and 21 days, when changes in FACS profiles were visible, the majority of cells are dead (as expected and as determined in Figure 7) and this could have caused loss of chromosomal integrity, potentially resulting in reduced fluorescence intensity. If an experimental validation of chromosomal integrity can’t be provided in short, the possibility of chromosomal degradation and the potential implication in FACS profiles could be mentioned and conclusions about ploidy changes softened a bit.

The way how the assumed ploidy reduction is mentioned in the abstract might lead to the misunderstanding that ploidy reduction was specific to heterozygous orc mutants, but in fact the changes in FACS profiles were also seen for the wild type, hence this effect is not connected to orc mutations. This should be clarified.

From the description of Figure 3 it does not become clear whether real time RT-PCR data has been normalized to a particular housekeeping gene. Please clarify.

A large part of the discussion (line 496-537) reviews background information on ORC without directs links to the results presented in the paper. This should be shortened and part of the information given could be moved to the introduction. The discussion should be more result oriented. It’s a puzzling finding that heterozygous state of orc1-6 increases CLS, while RLS is increased for all except orc6 mutants. It should be noted that other mutations are known which affect either RLS or CLS to make clear that both aging models are not always affected similarly. What is known about CLS/RLS of other DNA replication/repair mutants?

PMID: 29768403 studied morphological haploinsufficiency and observed this phenotype for heterozygous orc3 and orc6. This could be discussed.

Line 537: “Our data has clearly shown that the strains with the AAA+ domain (Orc1-Orc5) have significant growth disturbances and significant prolonged doubling times” - Strains lacking Orc proteins with an AAA+ domain have growth disturbances

Line 584: “but it was also showed” should read “was shown”

Line 585: “was proved” should read “was proven”

The text should be double checked for similar minor spelling issues

Author Response

Dear Professor

Re: Manuscript id: cells-1643187

Please find attached a revised version of our manuscript entitled "Depletion of the origin recognition complex subunits delays aging in the budding yeast", which we would like to resubmit for publication as a research paper in Cells.

We appreciate all the comments on the manuscript. According to suggestion, the manuscript has been edited by an English native speaker.

We are also grateful for constructive comments. We generally agree with the Reviewers' comments and list the point-to-point replies for easy reference. The manuscript has been greatly improved following the Reviewers' suggestions. We hope that you will find it worth publication in the Cells at the present shape.

We are looking forward to your decision.

Yours sincerely,
Mateusz Mołoń
Adrianna Skoneczna

Reviewer 3

  1. The manuscript provides an extensive phenotypic analysis of yeast strains heterozygous for ORC1-6, genes encoding origin recognition complex proteins. Interesting results about changes in two yeast specific models of aging, chronological lifespan (CLS) and replicative lifespan (RLS) were obtained.

Points to consider for improvement/clarification:

A potential ploidy loss of diploid yeast strains during chronological aging was observed and it is assumed that this effect is not the result of sporulation, which could be induced due to prolonged starvation and would give rise to haploid progeny. A sporulation defective strain could be used to clarify whether sporulation  is involved in ploidy loss during chronological aging. This could either be done experimentally or the possibility mentioned in the discussion/outlook.

Conclusions about ploidy reduction during chronological aging are based on flow cytometry detecting DNA fluorescence. To exclude the possibility that DNA degradation in the dead cell fraction has caused a down-shift in fluorescence intensity, it would be informative to assess chromosomal integrity, for example by PFGE. After 14 and 21 days, when changes in FACS profiles were visible, the majority of cells are dead (as expected and as determined in Figure 7) and this could have caused loss of chromosomal integrity, potentially resulting in reduced fluorescence intensity. If an experimental validation of chromosomal integrity can’t be provided in short, the possibility of chromosomal degradation and the potential implication in FACS profiles could be mentioned and conclusions about ploidy changes softened a bit.

The way how the assumed ploidy reduction is mentioned in the abstract might lead to the misunderstanding that ploidy reduction was specific to heterozygous orc mutants, but in fact the changes in FACS profiles were also seen for the wild type, hence this effect is not connected to orc mutations. This should be clarified.

We are glad that, similarly to us, the Reviewer found inspiring our result concerning haploidization of yeast genome during chronological aging. We appreciate all Reviewer's suggestions on how we could experimentally approach revealing the mechanism of this haploidization. These suggestions will be helpful in further research. The present manuscript regards a different subject, namely the influence of decreased ORC complex components level on cellular aging. In the discussion, we analyzed the possible sources of haploidization since the result was so intriguing that it was impossible to left it without comment. However, we are aware that exploring this unknown scientific field will require a lot of work in the future. For the moment, we will utilize the Reviewer's suggestion to soften a bit our conclusions concerning possible sources of haploidization and include the alternative hypothesis.

We are grateful to the Reviewer for the comment concerning our overlooked mistake. Indeed, the haploidization concerns all strains and is not specific to low doses of Orc1-6 subunits. We have corrected it.

  1. From the description of Figure 3 it does not become clear whether real time RT-PCR data has been normalized to a particular housekeeping gene. Please clarify.

Relative expression of ORC genes was normalized to ACT1. We added appropriate information in the description of Figure 3 and the Materials and methods section.

  1. A large part of the discussion (line 496-537) reviews background information on ORC without directs links to the results presented in the paper. This should be shortened and part of the information given could be moved to the introduction. The discussion should be more result oriented. It’s a puzzling finding that heterozygous state of orc1-6 increases CLS, while RLS is increased for all except orc6 mutants. It should be noted that other mutations are known which affect either RLS or CLS to make clear that both aging models are not always affected similarly. What is known about CLS/RLS of other DNA replication/repair mutants?

PMID: 29768403 studied morphological haploinsufficiency and observed this phenotype for heterozygous orc3 and orc6. This could be discussed.

If only possible, we would prefer to leave the arrangement proposed in the original manuscript. We believe the quick reminder concerning the ORC complex functioning makes it easier to follow the discussion.

According to the Reviewer's recommendations, we replenished and improved the part of the discussion regarding the discrepancy in the RLS or CLS phenotypes of heterozygous ORC/orc strains:

Here we also confirm that both aging models, i.e. RLS and CLS, are not always affected similarly. Previous studies have found that overexpression of the SOD1 and SOD2 genes decreases budding lifespan, but also slows down chronological aging [20]. In turn, other studies showed that, e.g. increased respiration in the sch9Δ mutant is required for increasing CLS but not RLS [65]. It has been shown recently that haploinsufficiency, i.e., a dominant phenotype caused by a heterozygous loss-of-function mutation, is rare. Interestingly in yeast, haploinsufficiency phenotypes was observed for more than half of essential genes under optimal growth conditions. Moreover, 40% of the essential genes without noticeable phenotypes under optimal growth conditions exhibited haploinsufficiency under extreme growth conditions [66]. This may explain the absence of a direct ORC6/orc6D phenotype in the replication aging model and significant alterations in the cell response to cell stress conditions during CLS, for instance, starvation or acidification of the medium. Therefore, we strongly support the suggestion that single-cell phenotyping is a powerful approach, even in heterozygous conditions.

  1. Line 537: “Our data has clearly shown that the strains with the AAA+ domain (Orc1-Orc5) have significant growth disturbances and significant prolonged doubling times” - Strains lacking Orc proteins with an AAA+ domain have growth disturbances
    Line 584: “but it was also showed” should read “was shown”
    Line 585: “was proved” should read “was proven”

We changed this sentences as suggested by the Reviewer.

  1. The text should be double checked for similar minor spelling issues

According to your suggestion, the manuscript has been edited by an English native speaker.

Round 2

Reviewer 1 Report

The authors provided sufficient supporting data and conclusions that fit the current findings.

Reviewer 2 Report

The manuscript has been improved. I think it is acceptable in its present form.